# Exploring the Landscape of Anti-Inflammatory Trials: A Comprehensive Review of Strategies for Targeting Inflammation in Acute Myocardial Infraction

**DOI:** 10.3390/biomedicines12030701

**Published:** 2024-03-21

**Authors:** Andreas Mitsis, Michaela Kyriakou, Stefanos Sokratous, Georgia Karmioti, Michail Drakomathioulakis, Michael Myrianthefs, Antonios Ziakas, Stergios Tzikas, George Kassimis

**Affiliations:** 1Cardiology Department, Nicosia General Hospital, Nicosia 2029, Cyprus; andymits7@gmail.com (A.M.); michaelakyriakou95@yahoo.com (M.K.); stefanossokratous94@gmail.com (S.S.); georgiakarm@outlook.com (G.K.); bageragr@gmail.com (M.D.); myr.michael@shso.org.cy (M.M.); 2First Department of Cardiology, AHEPA University Hospital, Aristotle University of Thessaloniki, 54636 Thessaloniki, Greece; aziakas@auth.gr; 3Third Department of Cardiology, Aristotle University of Thessaloniki, 54636 Thessaloniki, Greece; 4Second Department of Cardiology, Aristotle University of Thessaloniki, 54642 Thessaloniki, Greece

**Keywords:** acute myocardial infarction, anti-inflammatory interventions, canakinumab, colchicine, coronary artery disease, IL-6, inflammation

## Abstract

The role of inflammation in the pathophysiology of acute myocardial infarction (AMI) is well established. In recognizing inflammation’s pivotal role in AMI, this manuscript systematically traces the historical studies spanning from early attempts to the present landscape. Several anti-inflammatory trials targeting inflammation in post-AMI have been performed, and this review includes the key trials, as well as examines their designs, patient demographics, and primary outcomes. Efficacies and challenges are analyzed, thereby shedding light on the translational implications of trial outcomes. This article also discusses emerging trends, ongoing research, and potential future directions in the field. Practical applications and implications for clinical practice are considered by providing a holistic view of the evolving landscape of anti-inflammatory interventions in the context of AMI.

## 1. Introduction

Acute myocardial infarction (AMI) stands as a primary contributor to mortality and morbidity in cardiovascular disease (CVD) [1]. AMI is characterized by a complex interplay of physiological responses, including inflammation. The significance of inflammation in AMI is underlined by its role in aggravating tissue damage, influencing plaque stability, and contributing to adverse cardiac remodeling [2,3]. Recognizing inflammation as a key player in the outcome of AMI has prompted a growing interest in anti-inflammatory interventions as potential therapeutic strategies [4]. 

Early attempts to address inflammation in the context of AMI were marked by a gradual understanding of the complex relationship between inflammation and cardiac events [5,6]. Initial interventions primarily focused on broad-spectrum, anti-inflammatory agents, and the aim was to mitigate the inflammatory response associated with AMI [7,8]. Over time, as our understanding of the inflammatory pathways has developed, interventions have evolved to target specific components of the inflammatory cascade [9], and there has been a shift toward more dedicated strategies, including the exploration of cytokine modulation, antiplatelet agents [10,11], and immune-modulating therapies [12]. The evolution of these interventions reflects a growing awareness of the multifaceted nature of inflammation in AMI, thereby leading to more refined and targeted approaches to address this crucial aspect of cardiovascular (CV) pathology [13]. 

This comprehensive review aims to explore various strategies targeting inflammation post-AMI by providing insights into their mechanisms and clinical outcomes, thereby emphasizing their importance in enhancing CV health. The scope encompasses an exploration of various agents, mechanisms, and clinical outcomes, thus shedding light on the evolving landscape of therapeutic interventions that are aimed at mitigating inflammation in the context of AMI.

## 2. Classification of Anti-Inflammatory Trials in AMI

Organizing and classifying anti-inflammatory trials in CVD can be approached systematically. Anti-inflammatory trials can be classified according to the targeted inflammatory pathway, according to the class of anti-inflammatory drugs under investigation, and according to the patient population, as well as according to the outcome measures or even the trial phase (Table 1). This methodical approach enhances a thorough comprehension of the landscape and provides systematic organization for researchers, clinicians, and policymakers engaged in AMI-related inflammation research.

From the pathophysiology viewpoint, the ideal approach is to categorize inflammatory trials based on the specific inflammatory pathways or mediators targeted. Based on this, anti-inflammatory strategies can be classified into two main groups: those targeting the central interleukin-1 (IL-1), tumor necrosis factor-a (TNF-α), and interleukin-6 (IL-6) inflammatory signaling pathways; and those that operate independently of it [14]. The central IL-6 pathway plays a crucial role in the inflammatory signaling involved in the onset and advancement of CVD [15]. IL-6 coordinates the recruitment of immune cells to the injured myocardium [16]. It promotes the production of acute-phase proteins, such as the C-reactive protein (CRP), which serve as markers of inflammation [17]. Additionally, IL-6 influences the differentiation and activation of immune cells, thereby contributing to the inflammatory environment in the infarcted tissue [18,19]. In the other inflammatory cascade during AMI, key molecules outside the IL-6 pathway include matrix metalloproteinase-9 (MMP-9) [20], phospholipase-2 (PLA2) [21], P-selectin, E-selectin, intercellular cell adhesion molecule-1 (ICAM-1) [22], growth differentiation factor-15 (GDF-15) [23,24], and p38-mitogen-activated protein kinase (MAPK) [25]. These molecules collectively contribute to the interplay responses observed in the outcome of AMI, showing the complex nature of the inflammatory cascade in CV pathology. Therefore, addressing these elements within the central IL-6 signaling pathway or alternative inflammatory signaling pathways has the potential to delay the advancement of CV and systemic inflammation, thereby improving CV prognosis [26].

The pathophysiological organization of the anti-inflammatory trials is actual and reasonable but increasing the complexity might create confusion. For the purposes of this review, we decided to present the available studies based on the class of anti-inflammatory drugs under investigation, and the studies were divided in two categories: specific-target anti-inflammatory agents and broad-spectrum, anti-inflammatory agents (Table 2 and Table 3, respectively). This review encompasses the late-phase trials meeting the specific following criteria: (1) those conducted on individuals with established CVD; (2) those comparing anti-inflammatory therapies to a placebo; and (3) those designed to not only evaluate CV clinical outcomes and reporting on CV events and/or infarct size, but also on the effect on specific inflammatory biomarkers. 

## 3. Studies with Specific-Target Anti-Inflammatory Agents 

### 3.1. CANTOS Study (Canakinumab Anti-Inflammatory Thrombosis Outcomes Study)

The canakinumab anti-inflammatory thrombosis outcome study (CANTOS) was a randomized, double-blind, and placebo-controlled trial involving stable patients with previous myocardial infarction (MI), and it was evaluated whether canakinumab could prevent recurrent vascular events in men and women who have a persistent proinflammatory response [27].

Canakinumab, an interleukin-1 beta (IL-1β) inhibitor, was approved for use in several rare, heritable pediatric conditions associated with IL-1β over-expression [28]. The CANTOS trial examined the efficacy of canakinumab in reducing the CV events among 10,061 patients with a history of MI and elevated high-sensitivity C-reactive proteins (hsCRPs). The primary endpoint included major adverse CV events (MACE)—a composite of non-fatal MI, stroke, and CV death. The trial demonstrated a significant reduction in MACE in the canakinumab group, with a hazard ratio (HR) of 0.85 [95% confidence interval (CI) 0.74–0.98] and a *p*-value below 0.05. This landmark trial highlighted the link between inflammation, as indicated by hsCRP levels, and CV risk, thereby paving the way for targeted anti-inflammatory therapies in CVD [29]. The CANTOS trial demonstrated that inflammation plays a treatable role in atherosclerosis. By pinpointing IL-1β as a viable therapeutic target, it is poised to stimulate additional clinical investigations and the development of anti-inflammatory agents for CV prevention [30].

### 3.2. VISTA-16 Trial (Vascular Inflammation Suppression to Treat Acute Coronary Syndrome for 16 Weeks)

The plasma level of secretory PLA2 is a known risk factor for CAD and is associated with adverse outcomes in patients with stable CAD and acute coronary syndromes (ACS) [31,32,33]. The VISTA-16 trial, initiated in 2006, set out to investigate the efficacy of varespladib, which is a potent anti-inflammatory agent, by interfering with arachidonic acid metabolism (inhibitor of secretory PLA2) in terms of suppressing the vascular inflammation among patients with ACS. This randomized, double-blind trial enrolled 5145 participants and aimed to evaluate the impact of varespladib on MACE [34].

The trial was prematurely terminated in 2012 due to futility, as varespladib failed to demonstrate a statistically significant reduction in CV events compared to the placebo group (6.1% vs. 5.1%; HR, 1.25; 95% CI, 0.97–1.61; and *p*-value = 0.08). Of note, the composite secondary outcome of CV mortality, MI, and stroke was higher in the varespladib arm (4.6% vs. 3.8%, HR, 1.36; 95% CI, 1.02–1.82; *p*-value = 0.04). This was due primarily to a greater incidence of MI in the varespladib arm (3.4% vs. 2.2%; *p* = 0.005) [35]. The absence of a meaningful clinical benefit, coupled with the trial’s termination, underscored the limitations and challenges associated with targeting the secretory PLA2 as a therapeutic approach in ACS [36]. The findings prompted a critical reevaluation of the potential role of this pathway in ACS treatment strategies.

### 3.3. LATITUDE-TIMI 60 Trial (Losmapimod to Inhibit p38 MAP Kinase as a Therapeutic Target and Modify Outcomes after an Acute Coronary Syndrome)

Losmapimod is a selective, reversible, and competitive inhibitor of p38 MAPK [37]. MAPK-mediated inflammatory augmentation has been implicated in atherogenesis, plaque destabilization, and the detrimental processes in infarction and recovery [38], and it is considered an alternative inflammatory signaling pathway that acts outside the IL-6 pathway [39,40]. Preliminary data have shown that the use of losmapimod in non ST elevation MI (NSTEMI) patients attenuates inflammation and may improve outcomes [41]. 

The LATITUDE-TIMI 60 trial, a pivotal multinational investigation that delved into the efficacy and safety of losmapimod, was administered at a dosage of 7.5 mg twice daily in a cohort of 3503 patients that were presenting with ACS [42]. Contrary to expectations, the trial did not reveal a significant reduction in MACEs, including CV death, MI, and stroke (8.1% vs. 7.0%, HR, 1.16; 95% CI, 0.91–1.47; *p*-value  =  0.24) [42]. Losmapimod did not demonstrate a risk reduction for recurrent MACE events over the 12-week treatment period in patients hospitalized with ACS. Additionally, there was no indication that losmapimod influenced the occurrence of secondary outcomes, including all-cause mortality. Consequently, this study does not endorse the adoption of a strategy involving p38 MAPK inhibition with losmapimod for patients admitted with MI [43].

### 3.4. SOLID-TIMI 52 Trial (Stabilization of Plaques Using Darapladib-Thrombolysis in Myocardial Infarction 52)

Lipoprotein-associated phospholipase A2 (Lp-PLA2) has been proposed as a potential causal factor in atherosclerosis development, and it has also been suggested to contribute to plaque instability via the pathways associated with inflammation [32,44]. Darapladib is an oral selective Lp-PLA2 inhibitor that reduces Lp-PLA2 activity in plasma5 and in atherosclerotic plaques [45]. The use of darapladib in stable coronary artery patients did not show a significant reduction in the primary composite endpoint [46].

The SOLID-TIMI 52 trial, which was initiated in 2010, was a randomized, double-blind, placebo-controlled, multicenter, and event-driven trial. The focus was on investigating the role of darapladib in stabilizing atherosclerotic plaques and reducing CV events [47]. Enrolling over 13,000 patients, the trial aimed to evaluate darapladib’s impact on MACEs, such as CV death, MI, and stroke. However, the trial did not meet its primary endpoint as darapladib did not significantly reduce the risk of MACE compared to placebo [48]. The findings of SOLID-TIMI 52 raised questions about the role of Lp-PLA2 inhibition as a therapeutic target and prompted a further exploration of alternative approaches to address CV risk in patients with chronic CAD.

### 3.5. ASSAIL-MI-Trial (Assessing the Effect of Anti-IL-6 Treatment in Myocardial Infarction)

IL-6 is the crucial pro-inflammatory cytokine that is upregulated during MI, and it affects both plaque destabilization and myocardial remodeling [49,50]. Tocilizumab, an IL-6 receptor antagonist, can attenuate the inflammatory response and primarily the PCI-related TnT release in NSTEMI patients [12]. The ASSAIL-MI trial, a randomized, double-blind, and placebo-controlled study, was conducted at three high-volume PCI centers in Norway, where it was designed to evaluate the effect of tocilizumab on myocardial salvage in patients with acute ST-elevation MI (STEMI) [51,52]. Eligible participants included patients admitted with STEMI within 6 h of symptom onset. In a 1:1 randomization, consenting patients received a single infusion of either 280 mg tocilizumab or placebo promptly. The primary endpoint, evaluated using magnetic resonance imaging after 3 to 7 days, was the myocardial salvage index [52].

Out of the total, 101 patients were randomized to receive tocilizumab, and 98 patients received the placebo. The tocilizumab group exhibited a larger myocardial salvage index compared to the placebo group, with an adjusted between-group difference of 5.6 (95% CI: 0.2 to 11.3) percentage points (*p*-value = 0.04). While microvascular obstruction was less extensive in the tocilizumab arm, there was no significant difference in the final infarct size between the tocilizumab and placebo arms (7.2% vs. 9.1% of myocardial volume, *p*-value = 0.08). Adverse events were evenly distributed across the treatment groups. In conclusion, tocilizumab demonstrated an increase in myocardial salvage among patients with acute STEMI in the ASSAIL-MI trial [52].

### 3.6. SELECT-ACS (Effects of the P-Selectin Antagonist Inclacumab on Myocardial Damage after Percutaneous Coronary Intervention for Non-ST-Elevation Myocardial Infarction)

The SELECT ACS trial aimed to assess the efficacy of inclacumab in reducing myocardial damage during percutaneous coronary intervention (PCI) in patients with NSTEMI [53]. P-selectin, an adhesion molecule involved in cellular interactions (particularly among endothelial cells, platelets, and leukocytes), served as the target [54]. Inclacumab, a recombinant monoclonal antibody against P-selectin, was evaluated for its potential anti-inflammatory, antithrombotic, and antiatherogenic properties [55].

In this randomized trial involving 544 NSTEMI patients scheduled for coronary angiography and possible ad hoc PCI, the participants received one pre-procedural infusion of inclacumab at either 5 or 20 mg/kg or a placebo. The primary endpoint, assessed in patients who underwent PCI and received the study medication with available efficacy data (*n* = 322), was the change in troponin I from the baseline at 16 and 24 h after PCI.

The results indicated that inclacumab at 20 mg/kg demonstrated a significant reduction in troponin I levels, with a placebo-adjusted geometric mean percent change of −24.4% at 24 h (*p*-value = 0.05) and −22.4% at 16 h (*p*-value = 0.07). Similar trends were observed in the peak troponin I levels and the area under the curve over 24 h. The creatine kinase–myocardial band also showed reductions with inclacumab at 20 mg/kg [53]. Notably, adverse events did not significantly differ between the groups. In conclusion, inclacumab appeared to effectively diminish the myocardial damage following PCI in patients with NSTEMI [56].

## 4. Studies with Broad-Spectrum Anti-Inflammatory Agents

### 4.1. COLCOT Trial (Colchicine Cardiovascular Outcomes Trial)

Published in 2019, the COLCOT trial investigated the use of colchicine in 4745 post-MI patients [57]. Colchicine has a broad cellular effect that includes the inhibition of tubulin polymerization and the alteration of leukocyte responsiveness [58]. The primary endpoint, a composite of CV death, MI, stroke, resuscitated cardiac arrest, and urgent hospitalization for angina leading to coronary revascularization, exhibited an HR of 0.77 (95% CI 0.61–0.96), with a statistically significant *p*-value of 0.02. In individuals who recently experienced an MI, the use of low-dose colchicine demonstrated efficacy in preventing MACEs when compared to a placebo. The primary benefit stemmed from a notable reduction in the occurrence of stroke and the need for urgent hospitalization due to unstable angina leading to revascularization. Notably, colchicine exhibited positive effects, particularly among patients with diabetes. The study drug was well tolerated and exhibited a similar incidence of infection and diarrhea compared to the placebo (9.7% vs. 8.9%, *p*-value = 0.35). Additionally, colchicine was deemed cost-effective. The observed advantages of colchicine were attributed to the anti-inflammatory properties inherent in the drug. This trial underscored the potential of colchicine in preventing major CV events in this high-risk population, thereby shedding light on a novel anti-inflammatory approach in post-MI care [59].

### 4.2. LoDoCo2 Trial (Low-Dose Colchicine after Myocardial Infarction) 

The LoDoCo trial was designed to determine whether colchicine—a broad-spectrum, anti-inflammatory agent—when administered at 0.5 mg/day could minimize the risk of CV events in patients with clinically stable coronary disease [60]. The study was not placebo-controlled, and the primary outcome was the composite incidence of ACS, out-of-hospital cardiac arrest, or non-cardioembolic ischemic stroke. A total of 532 patients were included in the study. The primary outcome occurred in 15 of 282 patients (5.3%) who received colchicine and 40 of 250 patients (16.0%) who were assigned no colchicine (hazard ratio: 0.33; 95% confidence interval [CI] 0.18 to 0.59; *p* < 0.001; and number needed to treat: 11) [60]. 

The larger LoDoCo2 trial aimed to confirm the result of the LoDoCo trial. It was conducted in 2020 and enrolled 5522 post-myocardial infarction participants, and it focused on assessing the impact of low-dose colchicine on MACEs [61]. The primary endpoint, a composite of CV death, MI, ischemic stroke, or urgent hospitalization for angina requiring revascularization, exhibited a hazard ratio of 0.69 (95% CI 0.50–0.96), with a statistically significant *p*-value of 0.02 [62]. These trials not only confirmed the efficacy of colchicine in reducing CV risk, but also highlighted the potential benefits of using a lower colchicine dose in post-MI patients [63].

### 4.3. COPS Trial (Colchicine in Patients with Acute Coronary Syndrome)

COPS was multicenter, randomized, double-blind, and placebo-controlled trial involving 17 hospitals in Australia that provide acute cardiac care service [64]. The study enrolled 795 participants with ACS and investigated the potential utility of colchicine. The patients were randomized to receive colchicine or placebo in addition to standard secondary prevention pharmacotherapy, and they were followed up with for a minimum of 12 months. The primary outcome—a composite of all-cause mortality, ACS, unplanned urgent revascularization, and noncardioembolic ischemic stroke—did not significantly differ between the colchicine and placebo groups at 12 months (24 vs. 38 events and *p*-value = 0.09). Of note, the colchicine group showed a higher rate of total death, particularly non-CV death (*p*-value= 0.024). Adverse effects were similar between groups. In conclusion, colchicine did not significantly impact CV outcomes and was associated with a higher mortality rate in ACS patients.

### 4.4. CIRT Trial (Cardiovascular Inflammation Reduction Trial)

The Cardiovascular Inflammation Reduction Trial (CIRT), initiated in 2019, investigated the impact of low-dose methotrexate on CV events among individuals with a history of MI or multi-vessel CAD, and either type 2 diabetes or metabolic syndrome [65]. Enrolling a sizable population of 4786 patients, the trial aimed to discern the potential benefits of methotrexate in reducing MACE for a median follow up period of 2.3 years. The primary end point at the onset of the trial was a composite of nonfatal MI, nonfatal stroke, or CV death, as well as hospitalization due to unstable angina. Methotrexate did not result in lower IL-1β, IL-6, or CRP levels than placebo. The final primary end point occurred in 201 patients in the methotrexate group and in 207 in the placebo group (4.13 vs. 4.31 per 100 person-years; HR, 0.96; 95% CI, and 0.79 to 1.16). The original primary end point occurred in 170 patients in the methotrexate group and in 167 in the placebo group (3.46 vs. 3.43 per 100 person-years; hazard ratio, 1.01; 95% CI, and 0.82 to 1.25) [66].

### 4.5. AIM-HIGH (Atherothrombosis Intervention in Metabolic Syndrome with Low HDL/High Triglycerides: Impact on Global Health Outcomes)

Conducted in 2011, the AIM-HIGH trial assessed extended-release niacin in 3414 patients with a history of CVD and dyslipidemia. The trial did not demonstrate additional CV benefit from niacin therapy compared to placebo. The hazard ratio for the primary composite endpoint of cardiovascular events was 1.02 (95% CI 0.87–1.21), with a non-significant *p*-value of 0.79 [67]. AIM-HIGH raised questions about the efficacy of niacin in improving cardiovascular outcomes in this specific patient population, influencing subsequent considerations regarding niacin therapy in CV care [68].

### 4.6. ALL-Heart Study (Allopurinol versus Usual Care in UK Patients with Ischemic Heart Disease)

Elevated levels of serum uric acid have been linked to unfavorable CV outcomes [69]. While certain observational studies propose that therapy aimed at lowering uric acid may decrease CV risk [70], conflicting evidence exists as other studies have not observed similar benefits [71]. Allopurinol, a xanthine oxidase inhibitor, was approved for gout prophylaxis, symptomatic hyperuricemia treatment, and hyperuricemia prevention related to cancer chemotherapy. Allopurinol’s potential benefits for patients with ischemic heart disease may extend beyond its ability to lower serum uric acid levels. This includes its impact on decreasing the vascular oxidative stress mediated by xanthine oxidase, which could be a separate mechanism from its uric acid-lowering effects [72,73]. 

The ALL-HEART trial, a multicenter, prospective, and randomized study, aimed to investigate the impact of allopurinol therapy in patients aged 60 years or older with ischemic heart disease and no history of gout. The primary outcome, a composite of non-fatal myocardial infarction, non-fatal stroke, or CV death, showed no significant difference between the allopurinol and usual care groups. Over a mean follow-up of 4.8 years, the rates of the primary endpoint and overall mortality were similar between the groups, thus suggesting that allopurinol therapy does not confer additional CV benefits in this specific population [74].

## 5. Discussion

The studies reviewed above demonstrate that only CANTOS with canakinumab—as well as COLCOT, and LoDoCo2 with colchicine—achieved favorable clinical outcomes. The smaller trials like ASSAIL-MI with tocilizumab and SELECT-ACS with inclacumab, which assessed myocardial damage using CMR or troponin levels, showed promising results but need larger controlled studies to fully examine their impact on clinical outcomes.

Conversely, the remaining randomized trials did not show evidence that anti-inflammatory therapies can alter the prognosis in patients with CVD. This emphasizes that not all anti-inflammatory treatments are equivalent, and the method by which inflammation is reduced probably determines whether a particular anti-inflammatory drug will lower CV events. Therefore, a thorough examination and analysis of each study’s design can yield valuable insights.

When comparing CANTOS and CIRT, significant differences may contribute to their divergent outcomes [29,66]. While both trials primarily enrolled patients already on statin therapy, the patients in CIRT study achieved better LDL-C control and fell below the current guideline target (<70 mg/dL). Moreover, CANTOS required participants to have hsCRP levels of ≥2 mg/L, while CIRT did not mandate elevated hsCRP levels. Consequently, baseline hsCRP values were higher in CANTOS, thereby indicating a population with greater residual inflammatory risk. The most notable contrast lies in how these trials addressed inflammation: while CANTOS directly targeted the IL-1β pathway, thereby reducing the downstream mediators IL-6 and hsCRP, CIRT lowered inflammation (lower WBC) without altering IL-1β, IL-6, or hsCRP. Hence, the efficacy of anti-inflammatory therapies may vary, with the mechanism of inflammation reduction likely determining their impact on CV events.

The extent of hsCRP decrease after a single dose of canakinumab could offer a straightforward clinical approach to pinpoint individuals who may gain the greatest advantage from ongoing treatment [75]. Extracting this discovery in the design of all the other trials could explain that the inability to show effectiveness could also be linked to the mechanism, thereby highlighting the importance of selecting the appropriate inflammatory target or drug when inhibiting inflammation in CVD.

The remarkable outcome from the CANTOS trial demonstrated the successful reduction in CV risk with canakinumab therapy, thus highlighting the crucial inflammatory targets likely concentrated within the IL-1β to IL-6 to the CRP pathway [76]. Theoretically, colchicine, the other effective broad anti-inflammatory medication, has the potential to irreversibly inhibit the NLRP3 inflammasome, which leads to neutrophil dysfunction. This mechanism results in decreased circulating levels of IL-1β, IL-6, and CRP [77]. However, the mechanism of the CV that benefits from colchicine needs further evaluation. 

Based on the LoDoCo2 and CLCOT trials, low-dose colchicine has been considered the ideal anti-inflammatory treatment in patients with stable CAD [78]. Of note, a recent meta-analysis included more than 11,550 patients from the studies COLCOT, COPS, LoDoCo, and LoDoCo2, which showed that, in the secondary prevention of CV events, augmented standard medical therapy with low-dose colchicine decreases the occurrence of major CV events, except for CV mortality, in comparison to standard medical therapy alone [79]. The decrease in inflammation could be a critical factor in the effectiveness of low-dose colchicine in reducing the risk of recurrent cardiovascular events post-MI. Regular monitoring of hs-CRP levels before and after colchicine treatment could be significant [80].

It is important to highlight that many other cardiovascular medications have been shown to have anti-inflammatory effects. These drugs, through their various mechanisms, contribute to the management of inflammation in the cardiovascular system and may provide additional benefits beyond their primary indications. Statins, such as atorvastatin and rosuvastatin, exert pleiotropic effects beyond cholesterol reduction, including anti-inflammatory properties [81]. Ticagrelor, an antiplatelet agent, also shows anti-inflammatory effects beyond its primary role, and it is potentially achieved via improvement of vascular function and myocardial perfusion [82]. Interestingly, the research comparing ticagrelor to clopidogrel has assessed the impact of ticagrelor on CRP levels, which is a marker of inflammation and hints at potential anti-inflammatory actions beyond its antithrombotic effects [83]. Similarly, angiotensin-converting enzyme (ACE) inhibitors like lisinopril and angiotensin II receptor blockers (ARBs), such as losartan exhibit anti-inflammatory properties, reduce inflammation in the cardiovascular system [84,85]. Metformin, commonly used in diabetes management, has been found to have anti-inflammatory effects, which improves endothelial function [86]. Finally, thiazolidinediones like pioglitazone, act on peroxisome proliferator-activated receptors (PPAR-gamma agonists), and this also demonstrates anti-inflammatory actions [87]. Additional research is required to investigate the potential prognostic significance of these medications based on their anti-inflammatory effects.

## 6. Future Directions and Emerging Trends

Looking into the future of inflammation-targeted strategies in CV health, the trajectory is being guided by several key elements. Ongoing research and upcoming trials are composed to unravel deeper insights into the complexities of inflammatory pathways, thereby providing a novel understanding of their role in CV dynamics [88,89]. As the scientific community strives to expand its knowledge base, innovative approaches are anticipated to take center stage. These may encompass precision medicine [90], advanced imaging techniques [91], and evolving pharmacotherapies designed to modulate inflammation with greater specificity [92]. Table 4 includes summaries of all the ongoing clinical studies of treatments targeting inflammation in the context of atherosclerosis and AMI.

The landscape is also marked by emerging trends that signify a paradigm shift in addressing inflammation-related CV risks. From harnessing the potential of artificial intelligence in data analysis to exploring the role of microbiota in CV health, these trends underscore the multidimensional nature of ongoing investigations [93,94]. In this direction, attention has been drawn toward inflamm-aging. Inflamm-aging, which is characterized by chronic low-grade inflammation during aging, has been linked to various age-related diseases [95]. This phenomenon involves the presence of systemic inflammatory mediators in elderly individuals, which can exacerbate immune system disturbances and contribute to the development of several persistent diseases. Understanding the molecular mechanisms underlying inflamm-aging is crucial for developing targeted therapeutic strategies to mitigate its impact on health in aging individuals [96]. Similarly, attention was called into early arterial aging. Early arterial aging was considered when arterial age was higher than the biological age. Many studies have correlated high-sensitivity C-reactive protein with early arterial aging [97,98]. Therefore, systemic inflammation via elevated serum CRP levels may be related to a higher baseline disability from cardiovascular events [99]. 

Additionally, the assessment of diet factors seems to be important as another inflammatory component. For this purpose, the use of the dietary inflammatory index (DII), which is a tool developed to assess the inflammatory potential of diets by quantifying the cumulative effect of various dietary components on inflammation, has been widely used to evaluate the relationship between dietary patterns and the prevalence of various health conditions, including hypertension, cancer, endometriosis, hyperuricemia, sarcopenia, and diabetes [100]. Recent research has shown that the DII is positively correlated with the all-cause mortality of CAD patients. The intake of a pro-inflammatory diet may increase mortality in CAD patients [101].

Similarly, the use of the Systemic Immune-Inflammation Index (SII), a novel biomarker that integrates various components of the immune system to provide insights into systemic inflammation and its implications in different clinical scenarios, could be important in the evaluation of the degree of inflammation in CAD individuals. SII has been shown that it can also be used as a prognostic indicator in different cardiovascular scenarios [102,103,104]. Its ability to integrate immune parameters and provide valuable prognostic insights underscores its potential as a valuable tool in clinical practice for risk stratification and treatment decision making.

These potential advancements not only hold implications for scientific understanding, but also bear a profound impact on future clinical practice. A shift toward personalized, targeted interventions may redefine treatment strategies, thereby offering more effective and tailored approaches to patients [105]. As these trends unfold, they are set to shape the landscape of CV care, thus paving the way for a new era in the prevention and management of CVD.

## 7. Implications for Clinical Practice

In considering the implications for clinical practice, the integration of anti-inflammatory strategies into the management of AMI emerges as a transformative avenue. The practical applications of these strategies, which is explored in this section, shed light on their potential to redefine how we approach AMI treatment. As we navigate the intricacies of AMI management, understanding the significance of incorporating anti-inflammatory interventions into existing protocols becomes paramount [106]. This involves not only recognizing the specific clinical scenarios where such strategies prove most effective, but also ensuring a seamless integration that aligns with the broader treatment landscape [107]. 

In this direction, the new 2021 guidelines on CVD prevention that was issued by the European Society of Cardiology have suggested the consideration of low-dose colchicine (0.5 mg once daily) for the secondary prevention of CVD, especially in cases where recurrent events persist despite optimal therapy [108].

Moreover, the discussion extends beyond general applications to delve into the realm of personalized medicine. Tailoring anti-inflammatory interventions to the unique characteristics of individual patients represents a paradigm shift in clinical practice. By acknowledging patient-specific considerations, such as comorbidities, genetic predispositions, and lifestyle factors, clinicians can optimize the efficacy of anti-inflammatory strategies while minimizing potential risks. As we advance, these considerations forge a path toward a more patient-centered approach in the realm of AMI management, thus promising not only improved outcomes, but also a more tailored and compassionate form of CV care.

The ideal AMI target might be a patient with a large inflammatory burden. Extensive MIs, which are characterized by significant ischemic damage, release higher levels of inflammatory mediators, thereby leading to a more pronounced acute inflammatory response. These patients with extensive MIs, and consequently elevated inflammatory burden, might experience greater advantages from early anti-inflammatory treatments. Thus, the selection of patients with STEMI rather than NSTEMI, or the selection of patients with elevated levels of cardiac troponins, might be a useful approach in terms of promptly identifying individuals with substantial AMI and identifying the most suitable surrogate markers for anti-inflammatory therapy [109]. Additionally, the use of available biomarkers of inflammation (e.g., CRP and IL-6) could further improve the identification of individuals with a large inflammatory burden post-AMI, who would benefit from anti-inflammatory strategies. Finally, the ideal patient would benefit by the administration of anti-inflammatory therapy at early stages after the acute event. Inflammatory response may be protective in the early stage of the myocardial infarction through stimulation of myocyte autophagy. Anti-inflammatory treatment that is administered early after coronary occlusion may have an adverse effect [110]. A treatment plan aimed at addressing the initial stages of excessive and harmful post-AMI inflammation could potentially restrict further myocardial damage.

## 8. Conclusions

In conclusion, through the meticulous examination of pivotal trials, we uncovered diverse approaches, which range from therapies targeting the central IL-6 pathway to broad-spectrum, anti-inflammatory interventions. Despite the undoubted complexities in trial designs, patient populations, and primary outcomes, an understanding emerges, thereby highlighting both successes and challenges in the pursuit of effective anti-inflammatory interventions post-AMI. As we navigate this evolving field, our review not only synthesizes existing knowledge, but also underscores the need for continued exploration and innovation. This comprehensive overview sets the stage for future research and clinical endeavors, thereby providing a foundation for refining anti-inflammatory strategies and improving CV outcomes in AMI patients.

## Figures and Tables

**Table 1 biomedicines-12-00701-t001:** Approaches to organize and classify anti-inflammatory trials.

Targeted Inflammatory Pathway	Class of Anti-Inflammatory Drug	Patient Population	Outcome Measures	Trial Phase	Trial Result
Studies targeting the IL-6 pathway	Studies with specific-target anti-inflammatory agents	ACS patients	Trials assessing the impact on infarct size through anti-inflammatory interventions.	Early-phase trials (trials assessing the safety and initial efficacy)	Positive result
Trials outside the IL-6 pathway	Studies with broad anti-inflammatory agents	Stable CAD patients	Trials measuring clinical endpoints like MACE reduction.	Late-phase trials (larger trials evaluating effectiveness in a broader population)	Negative result

ACS: acute coronary syndrome; CAD: coronary artery disease; IL-6: interleukine-6; and MACE: major adverse cardiovascular event.

**Table 2 biomedicines-12-00701-t002:** Summary of clinical studies with specific-target, anti-inflammatory agents.

Trial Name	Year	Intervention	Patient Population	Follow up Period	Population (Number)	Key Findings	Notable Features and Considerations
CANTOS (Canakinumab Anti-inflammatory Thrombosis Outcomes Study)	2017	Canakinumab (IL-1β inhibitor)	Patients with prior MI and elevated hsCRP	48 months	10,061	Reduction in recurrent cardiovascular events in patients receiving canakinumab.	Notable for targeting interleukin-1β and demonstrating a link between inflammation (hsCRP) and cardiovascular risk.
VISTA-16 (Vascular Inflammation Suppression to Treat Acute Coronary Syndrome for 16 Weeks)	2014	Varespladib (phospholipase A2 inhibitor)	ACS patients (47% STEMI, 38% NSTEMI, 15% UA)	16 weeks	5145	No significant reduction in major cardiovascular events with varespladib.	Failed to prove the benefit of varespladib in patients with recent ACS who were on atorvastatin.
LATITUDE-TIMI 60 trial (Losmapimod to Inhibit p38 MAP Kinase as a Therapeutic Target and Modify Outcomes After an Acute Coronary Syndrome)	2016	Losmapimod (p38 MAPK inhibitor)	ACS patients (25% STEMI, 75% NSTEMI)	24 weeks	3503	No reduction for recurrent MACEs events over the 12-week treatment period in patients hospitalized with ACS.	Failed to support a strategy of p38 MAPK inhibition with losmapimod in patients hospitalized with MI.
SOLID-TIMI 52 trial (Stabilization of plaques using Darapladib-Thrombolysis in Myocardial Infarction)	2014	Darapladib (lipoprotein-associated phospholipase A2 (Lp-PLA2) inhibitor)	ACS patients (45.2% STEMI, 42.7 NSTEMI, and 12.2% UA)	2.5 years median	13,026	Darapladib did not reduce the risk of recurrent major coronary events.	Failed to support the use of targeted Lp-PLA2 inhibition with darapladib in patients stabilized after an ACS event.
ASSAIL-MI (Assessing the effect of Anti-IL-6 treatment in Myocardial Infarction)	2021	Tocilizumab (IL-6 receptor antagonist)	STEMI patients admitted within 6 h	7 days	199	Tocilizumab increased the myocardial salvage index compared to placebo. No significant difference in the final infarct size (7.2% vs. 9.1%, *p* = 0.08).	Conducted at three high-volume PCI centers in Norway; the single infusion of 280 mg tocilizumab or placebo; and the primary endpoint: the myocardial salvage index measured by MRI after 3 to 7 days.
SELECT ACS (Effects of the P-Selectin Antagonist Inclacumab on Myocardial Damage After Percutaneous Coronary Intervention for Non-ST-Elevation Myocardial Infarction)	2013	Inclacumab (anti-P-selectin)	NSTEMI patients undergoing PCI	24 h for efficacy and 120 days for safety evaluations	544	Inclacumab at 20 mg/kg demonstrated a significant reduction in troponin I levels at 24 h (*p* = 0.05) and 16 h (*p* = 0.07) after PCI compared to placebo. Adverse events did not significantly differ.	The P-selectin antagonist inclacumab reduced myocardial damage after PCI in patients with NSTEMI.

ACS: acute coronary syndrome; hsCRP: high-sensitive C reactive protein; MI: myocardial infarction; NSTEMI: non-ST elevation MI; PCI: percutaneous coronary intervention; p38 MAPK: p38 mitogen-activated protein kinases; STEMI: ST elevation MI; and UA: unstable angina.

**Table 3 biomedicines-12-00701-t003:** Summary of the clinical studies with broad-spectrum, anti-inflammatory agents.

Trial Name	Year	Intervention	Patient Population	Follow up Period	Population (Number)	Key Findings	Notable Features and Considerations
COLCOT (Colchicine Cardiovascular Outcomes Trial)	2019	Colchicine	Patients post-MI	Median 22.6 months	4745	Reduction in cardiovascular events in patients receiving colchicine.	Diarrhea was reported in 9.7% of the patients in the colchicine group and in 8.9% of those in the placebo group (*p* = 0.35).
LoDoCo2 (Low-Dose Colchicine after Myocardial Infarction)	2020	Colchicine	Patients post-MI	Median 28.6 months	5522	Reduction in major cardiovascular events with low-dose colchicine.	Focused on evaluating the efficacy of a lower colchicine dose in cardiovascular event prevention.
COPS (Colchicine in Patients with Acute Coronary Syndromes)	2020	Colchicine	ACS	12 months	795	The addition of colchicine to standard medical therapy did not significantly affect cardiovascular outcomes at 12 months in patients with ACS.	Colchicine was associated with a higher rate of mortality.
CIRT (Cardiovascular Inflammation Reduction Trial)	2019	Methotrexate	History of MI or multi-vessel CAD, and type 2 DM and/or metabolic syndrome	Median 2.3 years	4786	Methotrexate did not affect the cardiovascular outcomes or plasma markers.	Methotrexate was associated with modest elevations in liver enzyme levels and reductions in leukocyte counts and hematocrit levels, as well as a higher incidence of non-basal cell skin cancers than placebo.
AIM-HIGH (Atherothrombosis Intervention in Metabolic Syndrome with Low HDL/High Triglycerides: Impact on Global Health Outcomes)	2011	Extended-Release Niacin	Patients with a history of cardiovascular disease	3 years	3414	The trial did not demonstrate additional cardiovascular benefit from niacin therapy.	Raised questions about the efficacy of niacin in improving cardiovascular outcomes in this patient population.
ALL-HEART study (Allopurinol versus usual care in UK patients with ischemic heart disease))		Allopurinol	Patients with a history of cardiovascular disease and without gout	Mean 4.8 years	5721	No significant difference in the primary outcome between the two groups	While allopurinol has benefits in other conditions like gout, its role in reducing cardiovascular events in patients with CAD without gout may be limited

ACS: acute coronary syndrome; CAD: coronary artery disease; DM: diabetes mellitus; and MI: myocardial infarction.

**Table 4 biomedicines-12-00701-t004:** Summary of the ongoing clinical studies of treatments targeting inflammation in the context of atherosclerosis and acute myocardial infarction.

Trial Name	Study Design	Intervention	Target	Patient Population	Population (Number)	Primary Outcome	Clinical Trials Identifier
ZEUS	Phase III, multicenter, double-blind, randomized, and placebo-controlled	Ziltivekimab	IL-6 blocking monoclonal antibody	Patients with CKD stage 3 to 4, known CAD, and a hs-CRP of >2 mg/L	6200	Time to first occurrence of MACE	NCT05021835
Lp(a) HORIZON	Phase III, multicenter, double-blind, randomized, and placebo-controlled	Pelacarsen	Antisense oligonucleotide targeting Apo(a)	Patients with established CVD and a Lp(a) of ≥70 mg/dL	7680	Time to first occurrence of expanded MACE in patients with a Lp(a) of ≥ 70 mg/dL or a Lp(a) of ≥ 90 mg/dL	NCT04023552
GOLDILOX	Phase IIB, multicenter, double-blind, randomized, and placebo-controlled	MEDI6570	LOX-1 receptor blocking monoclonal antibody	Patients aged ≥ 21 years with a history of MI and a hs-CRP of >1 mg/L	400	Change in non-calcified plaque volume measured by CTA	NCT04610892
anaRITA MI2	Phase II multicenter, double-blind, randomized, and placebo-controlled	Rituximab	B-cell depletion with CD20	Patients with STEMI	558	LVEF at 6 months with cardiac magnetic resonance	NCT05211401
PULSE-MI	Randomized, multicenter, double-blind, and placebo-controlled clinical trial	A methylprednisolone 250 mg IV in a prehospital setting	Ischemia-reperfusion injury prevention and wide anti-inflammatory effect	Patients with STEMI	400	Infarct size measured by late-gadolinium enhancement on CMR at 90 days	NCT05462730
IVORY	Phase II, randomized, double-blind, placebo-controlled, and a parallel group	Low dose IL-2	Induces expansion of regulatory T cells	Patients with ACS or UA who have a hsCRP of >2 mg/L	60	Change in vascular inflammation measured by mean TBRmax in the index of 18F-FDG PET/CT	NCT04241601

ACS: acute coronary syndrome; CAD: coronary artery disease; CKD: chronic kidney disease; CMR: cardiac magnetic resonance; CTA: computed tomography angiography; FDG PET/CT: fluorodeoxyglucose-positron emission tomography; hsCRP: high sensitive C reactive protein; IL-2: interleukin-2; IL-6: interleukin 6; IV: intravenous; Lp(a): lipoprotein (a); LVEF: left ventricular ejection fraction; MACE: major adverse cardiovascular event; MI: myocardial infarction; STEMI: ST elevation MI; TBR: target-to-blood pool ratio; UA: unstable angina.

## Data Availability

Not applicable.

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
