# Peer review of "Exploring the Landscape of Anti-Inflammatory Trials: A Comprehensive Review of Strategies for Targeting Inflammation in Acute Myocardial Infraction"

_biomedicines, 2024, doi:10.3390/biomedicines12030701_

Round 1

Reviewer 1 Report

Comments and Suggestions for Authors

I have found the paper very useful and comprehensive. The antiinflammation was properly described and the recent studies were adequately cited and discussed. I would like to ask the authors to add a figure to the introduction, which helps the reader to imagine the targets in the inflammation cascade and eventually the positive/neutral and negative effects of the particular drugs.

Comments on the Quality of English Language

good paper

Author Response

Thank you for the positive feedback. With regards to the figure, we apologize for the inconvenience, but our editing tools cannot create something suitable for the level of the journal. Furthermore, we don’t have any funding to recruit graphic designers to generate this Figure. However, we do believe that the structure of the manuscript, together with the correctly placed tables can help the reader to understand the inflammatory pathways and the effect of anti-inflammatory medications listed in the text.

Reviewer 2 Report

Comments and Suggestions for Authors

The present paper aimed to explore various strategies targeting inflammation post-AMI, providing insights into their mechanisms and clinical outcomes, emphasizing their importance in CV health.

A few changes are needed, as follows:

Table 1: It is not clear which the significance of “Trials evaluating the effects on specific inflammatory biomarkers.” Where does it belong? Please mention that studies targeting IL-1 and other pathways will be included in the next tables.

Discussion and Future Directions: Please add a few words about inflamm-aging, the Dietary Inflammatory Index, systemic immune-inflammation index, high sensitivity C reactive protein as a predictor of early vascular aging and cardiovascular events and the Pleiotropic Effects of Ticagrelor

Please also mention that, according to Wang et al. “inflammatory response may be protective in the early stage of the myocardial infarction through stimulation of myocyte autophagy. Anti-inflammatory treatment early after coronary occlusion may have an adverse effect.” (Wang X, et al. Inflammation, Autophagy, and Apoptosis After Myocardial Infarction. J Am Heart Assoc. 2018 Apr 21;7(9):e008024. doi: 10.1161/JAHA.117.008024).

Author Response

Reviewer 2

‘’The present paper aimed to explore various strategies targeting inflammation post-AMI, providing insights into their mechanisms and clinical outcomes, emphasizing their importance in CV health.

A few changes are needed, as follows:

Table 1: It is not clear which the significance of “Trials evaluating the effects on specific inflammatory biomarkers.” Where does it belong? Please mention that studies targeting IL-1 and other pathways will be included in the next tables.’’

Reply: Thank you for your comment. The cell “Trials evaluating the effects on specific inflammatory biomarkers” has been removed for better readability of the table.

‘’Discussion and Future Directions: Please add a few words about inflamm-aging, the Dietary Inflammatory Index, systemic immune-inflammation index, high sensitivity C reactive protein as a predictor of early vascular aging and cardiovascular events and the Pleiotropic Effects of Ticagrelor’’

Reply: Thank you for this comment. We have added some details based on your suggestions.

Lines 356-373

It is important to highlight that many other cardiovascular medications have been shown to have anti-inflammatory effect. These drugs, through their various mechanisms, contribute to the management of inflammation in the cardiovascular system and may provide additional benefits beyond their primary indications. Statins, such as atorvastatin and rosuvastatin, exert pleiotropic effects beyond cholesterol reduction, including anti-inflammatory properties [81]. Ticagrelor, an antiplatelet agent, also shows anti-inflammatory effects beyond its primary role potentially via improvement of vascular function and myocardial perfusion [82]. Interestingly, research comparing ticagrelor to clopidogrel has assessed the impact of ticagrelor on CRP levels, a marker of inflammation, hinting at potential anti-inflammatory actions beyond its antithrombotic effects [83]. Similarly, angiotensin-converting enzyme (ACE) inhibitors like lisinopril and angiotensin II receptor blockers (ARBs) such as losartan exhibit anti-inflammatory properties, reducing inflammation in the cardiovascular system[84,85]. Metformin, commonly used in diabetes management, has been found to have anti-inflammatory effects, improving endothelial function[86]. Finally, thiazolidinediones like pioglitazone, act on peroxisome proliferator-activated receptors (PPAR-gamma agonists), also demonstrate anti-inflammatory actions [87]. Additional research is required to investigate the potential prognostic significance of these medications based on their anti-inflammatory effects.

Lines 396-408

In this direction, attention has been drawn into inflamm-aging. Inflamm-aging is characterized by chronic low-grade inflammation during aging, has been linked to various age-related diseases [95]. This phenomenon involves the presence of systemic inflammatory mediators in elderly individuals, which can exacerbate immune system disturbances and contribute to the development of several persistent diseases. Understanding the molecular mechanisms underlying inflamm-aging is crucial for developing targeted therapeutic strategies to mitigate its impact on health in aging individuals [96]. Similarly, attention was called into early arterial aging. Early arterial aging was considered when arterial age was higher than the biological age. Many studies have correlated high-sensitivity C-reactive protein with early arterial aging [97,98]. Therefore, systemic inflammation via elevated serum CRP levels may related with higher baseline disability from cardiovascular events [99].

Lines 410-418                                 

Additionally, assessment of diet factors seems to be important as another inflammatory component. For this purpose, the use of the dietary inflammatory index (DII), a tool developed to assess the inflammatory potential of diets by quantifying the cumulative effect of various dietary components on inflammation, has been widely used to evaluate the relationship between dietary patterns and the prevalence of various health conditions, including hypertension, cancer, endometriosis, hyperuricemia, sarcopenia, and diabetes[100]. Recent research showed that DII was positively correlated with the all-cause mortality of CAD patients. The intake of a pro-inflammatory diet may increase mortality in CAD patients [101].

Lines 419-426

Similarly, the use of the Systemic Immune-Inflammation Index (SII), a novel biomarker that integrates various components of the immune system to provide insights into systemic inflammation and its implications in different clinical scenarios, could be important in the evaluation of the degree of inflammation in CAD individuals. SII has been shown that it can also be used as a prognostic indicator in different cardiovascular scenarios [102–104]. Its ability to integrate immune parameters and provide valuable prognostic insights underscores its potential as a valuable tool in clinical practice for risk stratification and treatment decision-making.

‘’Please also mention that, according to Wang et al. “inflammatory response may be protective in the early stage of the myocardial infarction through stimulation of myocyte autophagy. Anti-inflammatory treatment early after coronary occlusion may have an adverse effect.” (Wang X, et al. Inflammation, Autophagy, and Apoptosis After Myocardial Infarction. J Am Heart Assoc. 2018 Apr 21;7(9):e008024. doi: 10.1161/JAHA.117.008024).’’

Reply: Done (see lines 468 – 470)